# *MEPE* loss-of-function variant associates with decreased bone mineral density and increased fracture risk

Ida Surakka[1,15], Lars G. Fritsche [1,2,15], Wei Zhou [3,4], Joshua Backman[5], Jack A. Kosmicki[5], Haocheng Lu [1], Ben Brumpton[6,7,8], Jonas B. Nielsen [1], Maiken E. Gabrielsen[6], Anne Heidi Skogholt[6], Brooke Wolford [4], Sarah E. Graham [1], Y. Eugene Chen[1], Seunggeun Lee [2], Hyun Min Kang[2], Arnulf Langhammer[9], Siri Forsmo[9], Bjørn O. Åsvold [6,9,10], Unnur Styrkarsdottir [11], Hilma Holm [11], Daniel Gudbjartsson [11,12], Kari Stefansson[11,13], Aris Baras [5], Regeneron Genetics Center[5,*], Goncalo R. Abecasis[2,5], Kristian Hveem[6,9 ✉] & Cristen J. Willer [1,4,6,14 ✉]

A major challenge in genetic association studies is that most associated variants fall in the non-coding part of the human genome. We searched for variants associated with bone mineral density (BMD) after enriching the discovery cohort for loss-of-function (LoF) mutations by sequencing a subset of the Nord-Trøndelag Health Study, followed by imputation in the remaining sample ($N = 19{,}705$), and identified ten known BMD loci. However, one previously unreported variant, LoF mutation in *MEPE*, p.(Lys70IlefsTer26, minor allele frequency [MAF] = 0.8%), was associated with decreased ultradistal forearm BMD ($P$-value $= 2.1 \times 10^{-18}$), and increased osteoporosis ($P$-value $= 4.2 \times 10^{-5}$) and fracture risk ($P$-value $= 1.6 \times 10^{-5}$). The *MEPE* LoF association with BMD and fractures was further evaluated in 279,435 UK (MAF = 0.05%, heel bone estimated BMD $P$-value $= 1.2 \times 10^{-16}$, any fracture $P$-value = 0.05) and 375,984 Icelandic samples (MAF = 0.03%, arm BMD $P$-value = 0.12, forearm fracture $P$-value = 0.005). Screening for the *MEPE* LoF mutations before adulthood could potentially prevent osteoporosis and fractures due to the lifelong effect on BMD observed in the study. A key implication for precision medicine is that high-impact functional variants missing from the publicly available cosmopolitan panels could be clinically more relevant than polygenic risk scores.

[1] Division of Cardiovascular Medicine, Department of Internal Medicine, University of Michigan, 1500 E. Medical Center Dr., Ann Arbor, MI 48109, USA. [2] Department of Biostatistics and Center for Statistical Genetics, University of Michigan School of Public Health, 1415 Washington Heights, 1700 SPH I, Ann Arbor, MI 48109, USA. [3] Program in Medical and Population Genetics, Broad Institute of Harvard and MIT, 415 Main Street, Cambridge, MA 02142, USA. [4] Department of Computational Medicine and Bioinformatics, University of Michigan, Palmer Commons, 100 Washtenaw Avenue, Ann Arbor, MI 48109, USA. [5] Regeneron Genetics Center, 777 Old Saw Mill River Road, Tarrytown, NY 10591, USA. [6] K.G. Jebsen Center for Genetic Epidemiology, Department of Public Health and Nursing, NTNU, Norwegian University of Science and Technology, NO-7491 Trondheim, Norway. [7] MRC Integrative Epidemiology Unit, University of Bristol, Oakfield House, Oakfield Grove, Bristol BS8 2BN, UK. [8] Clinic of Thoracic and Occupational Medicine, St. Olavs Hospital, Trondheim University Hospital, Prinsesse Kristinas gate 3, 7030 Trondheim, Norway. [9] HUNT Research Centre, Department of Public Health and Nursing, Norwegian University of Science and Technology, Postboks 8905, N-7491 Levanger, Norway. [10] Department of Endocrinology, St. Olavs Hospital, Trondheim University Hospital, Prinsesse Kristinas gate 3, 7030 Trondheim, Norway. [11] deCODE genetics/Amgen, Inc., Sturlugata 8, 101 Reykjavik, Iceland. [12] School of Engineering and Natural Sciences, University of Iceland, Sturlugata 7, 101 Reykjavik, Iceland. [13] Faculty of Medicine, University of Iceland, Vatnsmýrarvegur 16, 101 Reykjavik, Iceland. [14] Department of Human Genetics, University of Michigan, 4909 Buhl Building, 1241 E. Catherine St, Ann Arbor, MI 48109, USA. [15] These authors contributed equally: Ida Surakka, Lars G. Fritsche. *A list of authors and their affiliations appears at the end of the paper. ✉email: kristian.hveem@ntnu.no; cristen@umich.edu

The mineral content of bone reaches peak during young adulthood; as humans age, the mineral content of bone decreases and porosity increases, weakening the bones and leaving them vulnerable to fracture. Measurements of the density of bones, typically determined by x-ray absorption, can predict which individuals are at risk of hip, vertebral and other fractures but are often performed clinically only after a fracture occurs[1]. Bisphosphonate and other oral, subcutaneous and intravenous medications can be prescribed to increase bone mineral density (BMD) and reduce fracture risk in osteoporotic individuals[2]. BMD is typically measured in individuals with high risk of osteoporosis (for example, due to family history of osteoporosis, use of corticosteroids or use of antiestrogen in breast cancer treatment) and is recommended to be tested after a fracture.

BMD is a complex trait with a strong genetic component; heritability estimates range between 50 and 85%[3–6] and genome-wide association studies demonstrate this is mostly through polygenic effects. Additionally, there are multiple forms of monogenic skeletal diseases caused by single mutations[7], but these variants are typically very rare (<1/1000). Genome-wide studies of estimated BMD, as measured by ultrasound of heel have identified almost 900 associated genomic regions[8–10], with a substantial number also associated with fracture. Genomic discovery can aid in identifying targets for novel therapeutics, and potentially for identification of individuals at-risk for fracture that may benefit from preventive therapies.

Human diseases have typically been studied by testing associations between human genetic variation and phenotypes, where the discovered variants and genes are often investigated experimentally in model organisms or cell-based systems that can be genome-edited or perturbed in the laboratory. On the one hand, studying genetic mutations in humans themselves provides the natural genetic background and environmental conditions that lead to disease, but we are limited to observing the genetic changes that have arisen spontaneously in the human genome over time, and the frequency spectrum of variants that can be tested is limited by technology, cost and presence of those variants in the population under study. While on the other hand, the study of animal models can often provide conflicting or uninterpretable results when applied to humans, sometimes resulting in expensive clinical trials that fail.

To advance the translation of genetic discovery to improved therapeutics and prevention via prediction of at-risk individuals, we sought to identify rare and low-frequency loss-of-function (LoF) variants associated with BMD and fractures through a genome-wide association study (GWAS). We employed methodology wherein we performed low-pass sequencing of a subset of the sample ($N = 2202$), then imputed variants, including insertion/deletion polymorphisms, into the remainder of the HUNT discovery sample ($N = 19,705$) followed by replication of previously unreported variants in two independent replication samples: UK Biobank ($N = 279,435$) and deCODE ($N = 170,000$). Using this approach, we identify a LoF mutation in *MEPE*, which may be useful for precision medicine and therapeutic development.

## Results

### Genome-wide screen for BMD-associated LoF variants.

The Nord-Trøndelag Health Study (HUNT)[11] performed screening of BMD during enrollment into the HUNT study at different time points: HUNT2 in 1995–1997 and HUNT3 in 2006–2008. The standard technology in use at that time was single-energy x-ray absorptiometry (SXA), and the decision was made to focus on ultradistal forearm BMD measurements. Although this is not the current standard used in clinic or hospital-based cohort collections, the HUNT study has the advantage of a population-based screening of individuals with a wide variety of ages, with the inclusion of healthy individuals relative to a clinic-based phenotype, and decades of longitudinal clinical follow-up including fractures. Furthermore, it has been demonstrated that the T-score derived from wrist BMD is correlated with hip T-score ($r = 0.61$) and lumbar T-score ($r = 0.53$)[12], suggesting that ultradistal forearm BMD is helpful for estimating risk of fractures as well as diagnosing osteoporosis[13].

To enrich the discovery cohort for rare loss-of-function (LoF) variants typically missed by array-based genotyping, we first performed low-pass whole genome sequencing ($N = 2202$, on average 5X coverage) followed by imputation into the remaining HUNT samples, and tested for association with 11.2 M single nucleotide variants and 430,000 indels with high imputation quality (imputation $R^2 > 0.9$) and minor allele count >10 in 19,705 samples. We replicated 10 previously identified BMD loci with genome-wide significant associations with ultradistal forearm BMD (association test $P$-value $< 5 \times 10^{-8}$, Fig. 1, Table 1). One of the BMD-associated loci, *MEPE* on chromosome 4, spanned over a 5 Mb window and contained the lead intergenic variant reported for association with femoral neck and lumbar spine BMD by the GEFOS consortium in 2009 (rs1471403[14]) as well as the lead variant from the UK Biobank estimated heel bone mineral density (eBMD) GWAS[10] (rs11934731; $r^2 = 0.71$ with GEFOS lead variant). In the HUNT discovery cohort ($N = 19,705$), the minor allele at the lead single nucleotide variant at this locus, rs181831514, had a much higher impact (effect $= -0.53$ SD [standard deviation units], minor allele frequency [MAF] $= 0.8\%$), was much less common than the previously identified lead variants and was in nearly perfect linkage disequilibrium ($r^2 = 0.999$) with a rare LoF indel in the *MEPE* gene (rs753138805, *MEPE* p.Lys70IlefsTer26, $P$-value $= 2.1 \times 10^{-18}$). The insertion/deletion polymorphism was only observed following imputation from HUNT low-pass sequences which included indel calling. The Haplotype Reference Consortium imputation panel (which contains 1254 HUNT low-pass sequences which we submitted) was able to impute the intronic proxy variant (rs181831514; imputation $R^2 = 0.99$) but the indel was not present. The 1000 Genomes reference panel, which does include indel calls does not have p.Lys70IlefsTer26 present; however, the proxy variant imputation quality (imputation $R^2$) from 1000 Genomes reference dataset was 0.9979.

### Statistical evidence for *MEPE* p.Lys70IlefsTer26.

*MEPE* p.Lys70IlefsTer26 is located only 65 kb away from the previously identified association lead variant from a previous GWAS for eBMD in white British individuals[8] (LD $r^2 = 0.06$). The nominally significant association signal at this lead variant (rs11934731, effect size for the minor allele $= 0.029$ SD, $P$-value $= 0.01$, MAF $= 33\%$) was slightly attenuated when we performed conditional analysis with *MEPE* p.Lys70IlefsTer26 as a covariate in the model (effect size$_{conditional} = 0.021$ SD, $P$-value$_{conditional} = 0.06$; Supplementary Fig. 1, Supplementary Data 1). Additionally, the association for the *MEPE* LoF variant remains highly significant when conditioning for the lead variant of the eBMD analysis (effect size$_{conditional} = -0.529$ SD, $P$-value$_{conditional} = 5.3 \times 10^{-18}$).

The LoF deletion demonstrated a very strong association with ultradistal forearm BMD (effect size $= -0.53$ SD, $N = 19,705$, $P$-value $= 2.1 \times 10^{-18}$, Table 2) and has an eight fold stronger impact on BMD than the common variant previously reported as associated with femoral neck and lumbar spine BMD by the GEFOS Consortium[14] at this locus (effect size $= -0.07$ SD, MAF $= 0.34$). We replicated this finding in exome sequence data from 279,435 UK Biobank individuals with estimated heel BMD

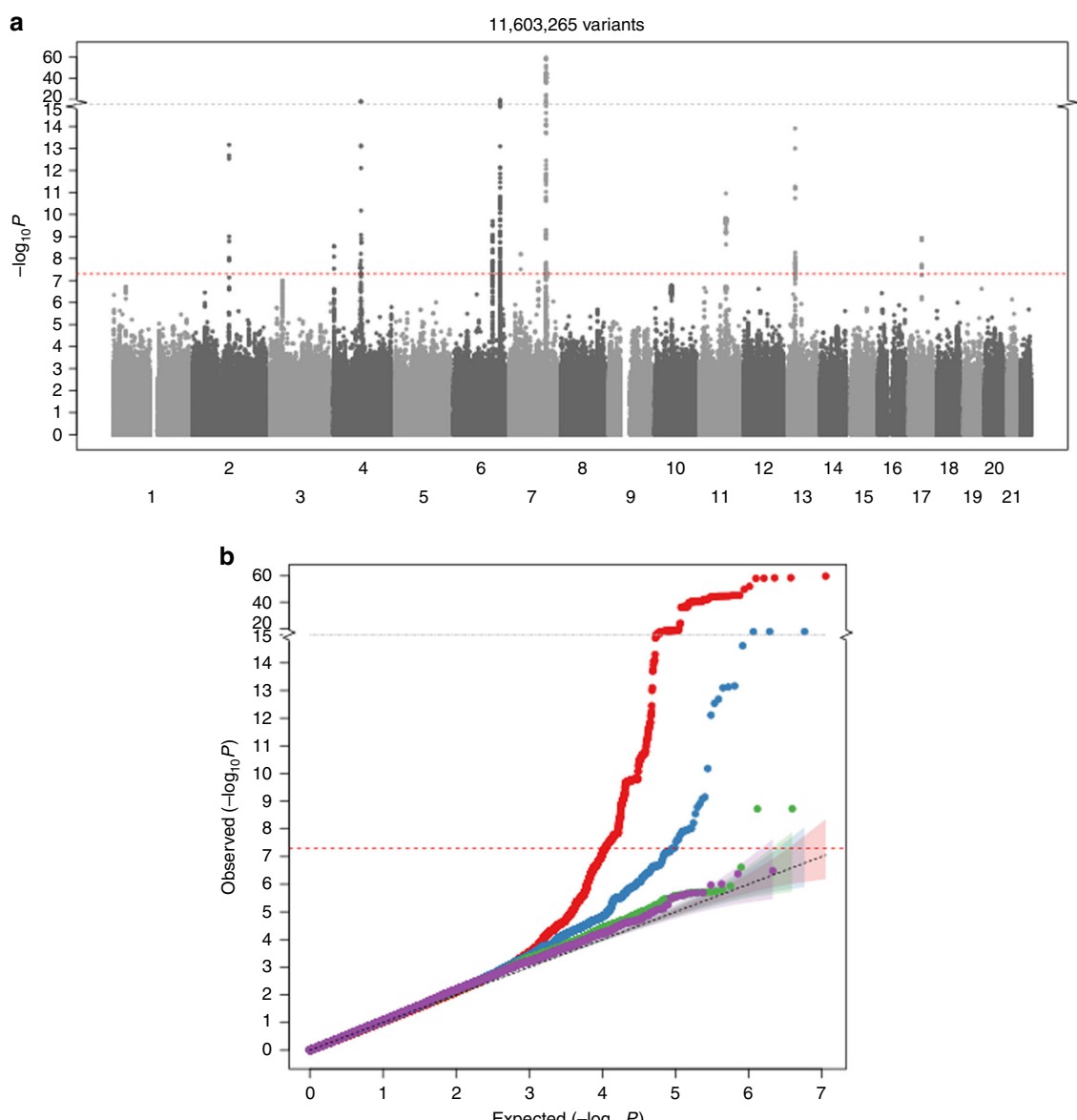

**Fig. 1 Ultradistal forearm BMD Genome-wide association analysis results.** Manhattan (upper panel) and QQ-plot (lower panel) in HUNT dataset ($N =$ 19,705) for ultradistal forearm bone mineral density (BMD) genome-wide association analysis. In the Manhattan plot (upper panel) the genome-wide significance threshold ($P$-value $< 5 \times 10^{-8}$) is shown using a red dotted line. In the QQ-plot (lower panel), the tested variants have been divided into four groups based on MAF (red dots = MAF [0.05; 0.5], blue dots = MAF [0.005; 0.05], green dots = MAF [0.001; 0.005], purple dots = MAF [0.000253; 0.001]). MAF: minor allele frequency. $-\log_{10}$P: $-1 \times$ tenth logarithm of the association test $P$-value.

(eBMD), which demonstrated a genome-wide significant association with a similar effect size (effect size $= -0.48$ SD, $N = 279,435$, $P$-value $= 1.2 \times 10^{-16}$, Table 2) as observed in Norwegian HUNT individuals. The proportion of UK individuals carrying at least one copy of *MEPE* p.Lys70IlefsTer26 was nearly 17 fold lower (0.095%) than the frequency observed in Norwegian HUNT individuals (1.6%). Additionally, this mutation shows comparable effect size (effect size for dual energy x-ray absorptiometry [DXA] arm BMD $= -0.588$, $N = 15,092$) in the Icelandic population where, however, the variant is even more rare (MAF $= 0.03$%), and thus lacks the power to replicate the association ($P$-value $= 0.12$).

**MEPE LoF variant clinical characterization.** To determine the impact of *MEPE* p.Lys70IlefsTer26 on clinical end points, we performed association analyses for bone-related phenotypes in HUNT (Supplementary Data 2) in the full HUNT dataset with genetic

information available ($N = 69,716$). *MEPE* p.Lys70IlefsTer26 carriers show higher risk for multiple types of fractures as well as osteoporosis with odds ratios (OR) ranging between 1.35 and 2.06 (Table 3). We see similar ORs in the Icelandic replication dataset, but the only significant association is for the forearm fractures of old individuals (Supplementary Table 1). In the UK population we see nominally significant association to any fracture (OR $= 1.76$ [1.00; 3.11], Supplementary Table 1).

Typical diagnostic criteria for osteoporosis is a BMD more than 2.5 SD below the reference population average (T-score $<$ $-2.5$). As we did not have the official T-score available for the dataset, which is always in relation to a reference dataset BMD distribution, we used the ultradistal forearm BMD lower than $-2.5$ SD as a proxy. Within the HUNT participants with genotypes and BMD measured ($N = 19,705$), the proportion of individuals who experienced any fracture during an average of 31

**Table 1 Genome-wide significant loci in the discovery HUNT sample.**

| rsID | Chromosome | Position (hg19) | Effect allele/ non-effect allele | Annotation (candidate gene) | Effect allele frequency | Imputation quality ($R^2$) | Effect size[a] (SE) | Association test $P$-value |
|---|---|---|---|---|---|---|---|---|
| rs115242848 | 2 | 119507607 | T/C | Intergenic (*EN1*) | 0.011 | 0.966 | 0.387 (0.052) | $6.9 \times 10^{-14}$ |
| rs4505759 | 4 | 1003022 | T/C | Upstream (*FGFRL1*) | 0.304 | 0.996 | 0.069 (0.012) | $2.8 \times 10^{-9}$ |
| rs181831514 | 4 | 88822746 | T/C | Intergenic (*MEPE*) | 0.008 | 0.988 | −0.533 (0.061) | $2.1 \times 10^{-18}$ |
| rs7741021 | 6 | 127468274 | C/A | Intronic (*RSPO3*) | 0.474 | 0.998 | 0.068 (0.011) | $2.0 \times 10^{-10}$ |
| rs4869742 | 6 | 151907748 | T/C | Intronic (*CCDC170*) | 0.273 | 0.992 | −0.108 (0.012) | $2.4 \times 10^{-19}$ |
| rs6973667 | 7 | 38152863 | G/A | Intergenic (*STARD3NL*) | 0.337 | 0.981 | 0.066 (0.011) | $6.2 \times 10^{-9}$ |
| rs2707518 | 7 | 120954908 | T/G | Intergenic (*WNT16*) | 0.367 | 0.989 | 0.182 (0.011) | $1.7 \times 10^{-60}$ |
| rs489247 | 11 | 86881641 | G/A | Intronic (*TMEM135*) | 0.258 | 0.997 | −0.083 (0.012) | $1.1 \times 10^{-11}$ |
| rs2147161 | 13 | 42982302 | C/A | Intergenic (*TNSFS11*) | 0.701 | 0.957 | −0.091 (0.012) | $1.2 \times 10^{-14}$ |
| rs76410205 | 17 | 41807508 | T/C | Intergenic (*SOST/DUSP3*) | 0.096 | 0.971 | 0.111 (0.018) | $1.2 \times 10^{-9}$ |

This table shows the 10 genome-wide significant ($P$-value $< 5 \times 10^{-8}$) loci associated to ultradistal forearm bone mineral density (BMD) in the discovery dataset ($N = 19,705$). As all these are previously known loci, the candidate gene has been taken from the previous publications. The effect of a variant is presented with the SAIGE linear mixed model effect size (Effect size) and standard error (SE) and the significance using the uncorrected two tailed Z-test $P$-value.
*rsID* reference SNP cluster ID, *SE* standard error of the effect estimate.
[a]Measured in SD units.

**Table 2 Association of the *MEPE* LoF variant to BMD phenotypes in the three study datasets.**

| Dataset | Frequency of the deletion allele | Phenotype | Effect for the deletion allele (SD units) | SE | N | Association test $P$-value |
|---|---|---|---|---|---|---|
| HUNT | 0.8% | Ultradistal forearm BMD | −0.53 | 0.061 | 19,705 | $2.1 \times 10^{-18}$ |
| UK Biobank | 0.05% | Estimated heel BMD | −0.48 | 0.059 | 279,435 | $1.2 \times 10^{-16}$ |
| deCODE | 0.03% | Whole body BMD | −0.62 | 0.372 | 14,194 | 0.10 |
| deCODE | 0.03% | Hip (femoral neck) BMD | −0.33 | 0.201 | 34,486 | 0.10 |
| deCODE | 0.03% | Arm BMD | −0.59 | 0.374 | 15,092 | 0.12 |
| deCODE | 0.03% | Lumbar Spine BMD | −0.07 | 0.214 | 33,746 | 0.76 |

This table shows the association results for all three datasets (HUNT, UK Biobank and deCODE) and all tested bone mineral density phenotypes for the *MEPE* loss-of-function frameshift deletion, p. Lys70IlefsTer26 (rs753138805, chr4: 88766219 GAAA/-). The effect of a variant is presented with the effect size (Effect size) and standard error (SE) and the significance using the uncorrected two tailed Z-test $P$-value.
*BMD* bone mineral density, *SD* standard deviation, *SE* standard error of the effect estimate, *N* number of samples.

**Table 3 Significant PheWAS results for the *MEPE* LoF mutation in HUNT dataset.**

| Description | OR [95% CI] | Association test $P$-value | #cases/#controls |
|---|---|---|---|
| Fracture of ankle and foot | 1.83 [1.42; 2.35] | $3.3 \times 10^{-6}$ | 5478/45480 |
| Fracture of upper limb | 1.51 [1.26; 1.82] | $1.2 \times 10^{-5}$ | 11128/45480 |
| Any fracture | 1.35 [1.18; 1.54] | $1.6 \times 10^{-5}$ | 24155/45480 |
| Fracture of radius and ulna | 1.61 [1.29; 2.00] | $1.8 \times 10^{-5}$ | 7998/45480 |
| Fracture of foot | 2.06 [1.48; 2.86] | $2.0 \times 10^{-5}$ | 3223/45480 |
| Osteoporosis | 1.58 [1.27; 1.97] | $4.2 \times 10^{-5}$ | 6994/61558 |
| Osteoporosis, osteopenia and pathological fracture | 1.50 [1.22; 1.84] | $1.1 \times 10^{-4}$ | 8077/61558 |
| Senile osteoporosis | 1.69 [1.28; 2.22] | $1.8 \times 10^{-4}$ | 4482/61558 |
| Fracture of humerus | 2.01 [1.39; 2.90] | $2.0 \times 10^{-4}$ | 2457/45480 |
| Fracture of unspecified bones | 1.46 [1.19; 1.79] | $3.4 \times 10^{-4}$ | 8627/45480 |
| Fracture of hand or wrist | 1.52 [1.19; 1.94] | $7.7 \times 10^{-4}$ | 5860/45480 |

This table presents all significant ($P$-value $< 1.2 \times 10^{-3}$, Bonferroni correction for 42 phenotypes) end-point associations for the *MEPE* LoF frameshift deletion in HUNT dataset ($N = 69,716$). The effect of a variant is presented with the odds ratio (OR) and 95% confidence intervals (CI) and the significance with uncorrected two tailed Z-test (for log(OR)) $P$-value. Full phenome-wide association scan (PheWAS) results and ICD codes underlying the phenotypes can be found from Supplementary Data 2.
*OR* odds ratio, *CI* confidence interval.

years of follow-up was 41.0% ($N = 8082$)—this includes all types of fractures and all causes including trauma. Within the relatively small subset of individuals with ultradistal forearm BMD $<$ −2.5 SD at the time of BMD measurement, 53.3% had experienced a fracture (65 of 122; Supplementary Table 2). Similarly, almost half (49.0%) of the individuals that carried the p.Lys70IlefsTer26 mutation had experienced a fracture (149 of 304, OR $= 1.39$ [1.11; 1.74]) during the follow-up period, which was not significantly different from those with low BMD (BMD $<$ −2.5 SD, OR $= 1.65$ [1.15; 2.35]).

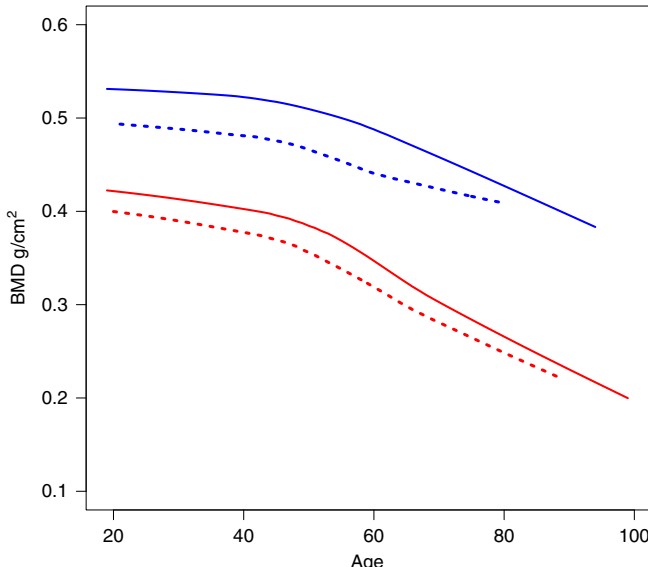

**Fig. 2 Age trend in BMD for _MEPE_ LoF mutation carriers.** In this figure we have compared the forearm bone mineral density (BMD) in the _MEPE_ loss-of-function (LoF) mutation, p.Lys70IlefsTer26, carriers (dotted lines) compared to non-carriers (solid lines) in the HUNT dataset ($N = 19,705$). The trend over different ages is illustrated using LOWESS curve for males (blue lines) and females (red lines) separately.

To compare the potential clinical impact of this single variant, we calculated a polygenic score for eBMD in the HUNT dataset based on Kim et al.[8] results using 1032 independently associated variants and weights from the UK Biobank cohort (Supplementary Data 3). Of the HUNT individuals in the lowest 1% of the BMD polygenic score distribution (i.e., 1% of the population with highest genetic burden for low BMD, fractures and osteoporosis), 38.2% experienced fracture during the follow-up, which is not significantly different from the rate in the remaining individuals (OR = 0.89 [0.66; 1.19], Fisher test _P_-value = 0.46, Supplementary Table 2). We performed the same comparison for the 5% and 10% tails, and similarly saw no difference in the fracture rates (OR = 0.98 [0.86; 1.12], Fisher test _P_-value = 0.79 and 1.04 [0.94; 1.14], Fisher test _P_-value = 0.44, respectively).

When examining the decrease in ultradistal forearm BMD by age in both LoF carriers and non-carriers, we can see that the loss of BMD in both subgroups is the same (Fig. 2) suggesting that _MEPE_ p.Lys70IlefsTer26 mutation affects the BMD peak value, rather than the lifetime bone mass loss similarly to the _LGR4_ stop-gain variant previously identified by Styrkarsdottir et al.[15]. In our dataset, the ultradistal forearm BMD of an average 20 year-old woman carrying the LoF mutation in _MEPE_ has a similar ultradistal forearm BMD as a 54 year-old non-carrier woman. A 20 year-old man carrying the LoF mutation in _MEPE_ had an ultradistal forearm BMD similar to the average BMD for a 64 year-old male carrier (Supplementary Fig. 2).

## Discussion

By sequencing followed by imputation into a large population-based study in Norway, we identified a LoF mutation in the _MEPE_ gene, p.Lys70IlefsTer26, that demonstrates a genome-wide significant and high-impact association with ultradistal forearm BMD and with increased risk of osteoporosis and fractures. Common variants at this locus have been associated with BMD-related traits in previous studies[10,14], but we were able to pinpoint this association to a rare LoF mutation in the gene, definitively

establishing a causal role and direction of effect of _MEPE_ on BMD.

Matrix extracellular phosphoglycoprotein (_MEPE_) was first cloned as a candidate for the oncogenic phosphaturic factor with similarities to a group of bone–tooth mineral matrix phospho-glycoproteins[16]. _MEPE_ is expressed mainly in bone marrow and brain bone-associated tumors[16]. Mice with an ablated _MEPE_ gene displayed increased osteoblast number, osteoblast activity and a higher bone mass[17], whereas _MEPE_ overexpression in bone inhibited bone growth and mineralization in mice[18]. Although how _MEPE_ regulates osteoblast activity remains unclear, _MEPE_ expression shows different patterns during osteoblast differentiation in human and mice. In mice, Mepe increases with differentiation[19], while in human _MEPE_ expression peaks in the proliferation phase and is suppressed during further differentiation[20].

_MEPE_ p.Lys70IlefsTer26 is the second rare LoF mutation found to be associated with BMD in population-based genome-wide association studies. A study by Styrkarsdottir et al.[15] found a rare stop-gain mutation in _LGR4_ gene with higher effect than the _MEPE_ frameshift mutation (effect size for whole body BMD = $-0.75$ SD, _P_-value $= 1.6 \times 10^{-6}$, $N = 7359$) that was associated with a binary phenotype for low hip, spine or whole body BMD (defined as $< -1.0$ SD, _P_-value $= 1.3 \times 10^{-10}$). The same research group has also identified two rare missense mutations in _COL1A2_ associated with low BMD[21] in an Icelandic dataset with $N = 209,379$ (p.Gly496Ala; MAF = 0.1%, effect size for low spine BMD $< -1$SD = 1.53, _P_-value $= 1.8 \times 10^{-7}$ and p.Gly703Ser; MAF = 0.05%, effect size for low hip BMD $< -1$SD = 2.23, _P_-value $= 1.9 \times 10^{-8}$). A study by Zheng et al.[22] identified non-coding mutations with MAF~1% in _EN1_ and _WNT16_ genes using whole-genome sequence data from the UK10K cohort ($N = 2882$), followed by imputation into over 20,000 samples. Their results demonstrated a 4-fold increase in the effect size for the low-frequency variants compared to the common variants found in the previous GWAS.

Because the mutation status can be determined at birth, _MEPE_ LoF carriers may benefit from treatment to preserve or increase peak BMD prior to the age of peak BMD. As the carriers show a similar rate of BMD loss during the adulthood as non-carriers, it may be sufficient for these individuals to be treated during the late childhood and early adulthood to increase the peak BMD. The promise of possible prevention of fractures by screening the population for _MEPE_ mutations relative to measuring BMD is that _MEPE_ p.Lys70IlefsTer26 carriers can be identified and pro-vided treatment prior to decrease of BMD, which has the potential to prevent fractures in this subgroup and perhaps maintain higher lifetime BMD. Also, _MEPE_ LoF carrier status identifies twice as many at-risk individuals as the BMD $< -2.5$ SD criterion in HUNT study participants (1.54% carry LoF vs 0.62% have BMD $< -2.5$ SD). We suggest that current clinical practice could be augmented with additional screening for the carriers of LoF variants of the _MEPE_ gene. The finding that carriers of the mutation had lower ultradistal forearm BMD even during young adulthood, when bone mass would be expected to peak, suggests that these individuals may benefit from early initiation of osteo-porosis prevention.

There are some limitations to our study. We acknowledge the difference between the discovery dataset and replication pheno-types, ultradistal forearm BMD measured with SXA (HUNT), compared to DXA from lumbar spine, hip, arm and whole body in deCODE and heel bone BMD measured with ultrasound in the UK biobank. However, as can be seen from Supplementary Fig. 3, the correlation between effect estimates for these two phenotypes is fairly high (0.7–0.8) when comparing SNPs with adequate power in the smaller dataset. Additionally, as we have hospital registry data from the Nord-Trøndelag county only, it is possible

that some of the HUNT study participants have experienced a fracture that is not accounted for in our dataset.

Mendelian randomization studies have demonstrated that BMD is a causal risk factor for fracture[23]. Therefore, we suggest that screening for individuals at high genetic risk could aid in starting appropriate pharmaceutical therapies and avoid fracture risk in these individuals. We performed low-pass whole genome sequencing of 2202 individuals followed by imputation into ~20,000 individuals from the HUNT study of Norway. We did not deeply sequence the *MEPE* gene in all Norwegians in this study, suggesting that additional LoF variants in this gene may be observed. By identifying other *MEPE* LoF mutations carriers, on top of the 1.6% with the p.Lys70IlefsTer26, we could increase the number of individuals who could be protected from fracture caused by low BMD. In addition to the European population, the *MEPE* LoF variant is present in African and Latino populations but with an extremely low allele frequency. However, given the presence of 16 different LoF mutations in the UK population[24], different LoF mutations may be present, but as-yet-undetected in other populations.

This study demonstrates that continued investigation of genetic variation in humans, particularly rare variants identified through sequencing, can identify genetic variants that clearly and immediately define functional genes and may be useful for precision medicine and therapeutic development.

## Methods

**HUNT genotype dataset**. The discovery dataset, The Nord-Trøndelag Health Study (HUNT)[11], is a population-based cohort of ~120,000 (descriptives in Supplementary Table 3) from the county of Nord-Trøndelag, Norway. Since 1984, phenotype data has been collected for these individuals approximately every 11 years. Participation in the HUNT Study is based on informed consent and the study has been approved by the Data Inspectorate and the Regional Ethics Committee for Medical Research in Norway (REK: 2014/144). In total, approximately 70,000 HUNT individuals have been genotyped using the Illumina Human CoreExome v1.1 array from both HUNT2 (collected between 1995–1997) and HUNT3 (collected between 2006 and 2008). After quality control of the genotype data, 69,716 European ancestry samples were imputed using a combined Haplotype Reference Consortium reference panel and a population specific whole genome sequence-based imputation panel[25]. 11.2 M single nucleotide variants and 430,000 indels with high imputation quality (imputation $R^2 > 0.9$) and minor allele count >10 were included in the analysis. Variants were annotated as a LoF mutation (3510 variants) if predicted as LoF (stop gain, stop loss, splice variant or frameshift) for either UCSC, Ensembl or RefSeq transcripts by ANNOVAR in Whole Genome Sequence Annotator[26] v0.7.

**HUNT phenotypes**. BMD (in g/cm$^2$) was measured at the ultradistal part of the non-dominant forearm by single-energy x-ray absorptiometry (DTX100; Osteometer MediTech, Inc, Hawthorne, CA), and the measurements were standardized using equipment-specific correction factors[27] estimated by three repeated hydroxyapatite bone imitation measurements of the European Forearm Phantom (QRM GmbH, Moehrendorf, Germany). BMD was measured in a subset of adult HUNT participants, including: 5% random samples of all participants, random samples of female participants in selected municipalities and age-groups within 35–85 years of age, and participants reporting ever having asthma, asthma-related symptoms or use of asthma medication (detailed selection criteria are available at ntnu.edu/hunt/databank). The present analyses were restricted to participants of European ancestry. For individuals who had their BMD measured in either HUNT2 collection, HUNT2 follow-up (2001) or HUNT3 collection, the HUNT2 measurement was prioritized, follow by HUNT3, then HUNT2 follow-up. The final discovery analysis dataset consisted of 19,705 samples with both imputed genome information and BMD.

Using the unique 11-digit national identification number that is allocated to all Norwegian citizens, we linked the HUNT study data to prospectively recorded information on fractures at the hospitals serving Nord-Trøndelag county: the local Levanger and Namsos Hospitals (Nord-Trøndelag Hospital Trust) and St. Olavs Hospital, Trondheim University Hospital. ICD-9 and ICD-10 codes from the electronic patient administrative systems were available from all hospitals from September, 1987 through October, 2017. For forearm and hip fractures at Levanger and Namsos Hospitals from October, 1995 through December, 2012, all diagnoses were validated by examination of medical records (relevant ICD codes accompanied by a procedure code for reduction, surgical intervention, or intervention with a rigid device), confirmation by X-ray or by review by a medical

doctor[28]. A full list of PheCodes (derived from ICD-9 and ICD-10 codes) included in the phenome-wide association analysis can be found in Supplementary Data 2.

**Statistical methods for discovery in HUNT**. The association analysis in the discovery dataset was performed using SAIGE[29], which implements linear or logistic mixed effects model (for quantitative and binary phenotypes respectively) accounting for sample relatedness and subtle population structure. The association analyses for inverse normal transformed ultradistal forearm BMD and clinical end points were adjusted with age (birth year for the clinical end points), sex, the first 4 genetic principal components and genotyping batch. Formal conditional analysis for the *MEPE* locus was performed using the same software, model and covariates as the discovery association analysis by adding the LoF variant as an additional covariate in the linear mixed model. Due to power restrictions, the analyses for clinical end points were restricted to PheCode-derived diagnoses with at least 500 cases. The sample size for the ultradistal forearm BMD association analysis was $N = 19,705$ and for the clinical end-point analyses $N = 69,716$. Clinical end points reaching $P$-value $< 1.2 \times 10^{-3}$ (Bonferroni correction for 42 end points) were regarded as statistically significant. The LOWESS (Locally Weighted Scatterplot Smoothing) curve for age trend was fitted using smoother span (the proportion of points in the figure affecting the local value) of 2/3. The Fisher tests for comparing different predictors for fractures and all the Figures have been done using R (https://cran.r-project.org) v3.5.3.

**Replication datasets**. Replication of the association at the *MEPE* LoF variant, p. Lys70IlefsTer26, was tested within the UK Biobank whole-exome sequence dataset in 279,435 participants. All participants in the UK Biobank provided informed consent and the study has obtained Research Tissue Bank (RTB) approval from its ethics committee (The Research Ethics Committee approval number; 11/NW/0382). Detailed cohort descriptions, sequencing, imputation and analysis methods for the UK biobank replication dataset can be found from Van Hout et al.[24]. Briefly, 302,342 participants (of which 279,435 with eBMD) were exome sequenced (coverage exceeds 20X at 95.5% of sites on average) resulting in ~12 million variants in targeted regions. Heel bone quality was evaluated with two methods; quantitative ultrasound speed of sound and broadband ultrasound attenuation using a Sahara Clinical Bone Sonometer (Hologic Corporation, Bedford, Massachusetts, USA)[10]. Raw values for eBMD were first stratified by sex, rank-inverse normal transformed, and then re-combined. Association analysis was performed using a linear mixed model implemented in BOLT-LMM v2.3.2 (https://data.broadinstitute.org/alkesgroup/BOLT-LMM/) with covariates for age, age-squared, and first ten genetic principal components. The replication analysis for fractures (ICD codes S22–S92, excluding skull [S02] and neck [S12]) was ran using SAIGE with age, age$^2$, age–sex interaction, sex and first ten genetic principal components as covariates.

The Icelandic replication dataset (deCODE)[30,31] is based on 170,000 genotyped samples which have been imputed using a whole-genome sequenced population specific imputation panel. All participating individuals, or their guardians, gave their informed consent before blood samples were drawn and the study has been approved by the National Bioethics Committee and the Icelandic Data Protection Authority. Using these samples, the genotypes of 375,984 samples have been imputed using familial imputation. The imputation quality for the *MEPE* p.Lys70IlefsTer26 (imputation info score) was 0.99. The BMD in the dataset has been measured using DXA from lumbar spine, hip, arm and whole body. Additionally, the dataset has health-care registry data available, which have been used in the end-point association replication.

## Data availability

The GWAS summary statistics are available at http://csg.sph.umich.edu/willer/public/bmd2020/. All other data that support the findings of this study are available from the corresponding author upon reasonable request. The Haplotype Reference Consortium imputation panel is accessible through the Michigan Imputation Server (https://imputationserver.sph.umich.edu/index.html#!). The UK Biobank replication cohort is a publicly available dataset for research purposes and can be accessed/applied from https://www.ukbiobank.ac.uk. The deCODE Genetics dataset summary results can be requested from the deCODE authors.

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

## Acknowledgements

The authors thank all HUNT, UK Biobank and deCODE genetics study participants for their contributions to research. The Nord-Trøndelag Health Study (The HUNT Study) is a collaboration between HUNT Research Centre (Faculty of Medicine and Health Sciences, NTNU, Norwegian University of Science and Technology), Nord-Trøndelag County Council, Central Norway Regional Health Authority, and the Norwegian Institute of Public Health. The K.G. Jebsen Center for Genetic Epidemiology is financed by Stiftelsen Kristian Gerhard Jebsen; Faculty of Medicine and Health Sciences, Norwegian University of Science and Technology (NTNU); and Central Norway Regional Health Authority. The replication in UK Biobank has been conducted using the data application number 26041. J.B.N. was supported by grants from the Danish Heart Foundation and the Lundbeck Foundation. Y.E.C was supported by NIH grants R01-HL068878 and R01-HL137214. C.J.W. was supported by NIH grants R35-HL135824, R01-HL127564, R01-HL117626-02-S1, and R01-HL130705. I.S. is supported by a Precision Health Scholars Award from the University of Michigan Medical School. Regeneron Genetics Center authors are listed in alphabetical order. Detailed author contributions for Regeneron Genetics Centre authors can be found from Supplementary Note.

## Author contributions

I.S., L.G.F., K.H., and C.J.W. designed the study. I.S., L.G.F., W.Z., H.L., B.B., J.B.N., B.W., and S.E.G. analyzed the data. H.L. and Y.E.C. evaluated functional follow-up experiments. L.G.F., J.B.N., M.E.G., A.H.S., A.L., S.F., B.O.Å. contributed to the phenotype harmonizations. J.B., J.A.K., U.S., H.H., D.G., K.S., A.B., Regeneron Genetics Center and G.R.A. contributed to the replications. W.Z., B.B., S.L., H.M.K., D.G., and G.R.A. provided statistical expertise. J.B.N., Y.E.C., A.L., S.F., B.O.Å., U.S., H.H., and K.H. provided clinical expertise. I.S., L.G.F., K.H., and C.J.W. wrote the paper. All the authors read and revised the manuscript.

## Competing interests

G.R.A., J.B. J.A.K., and A.B. work at Regeneron. U.T., H.H., D.G., and K.S. are employed at deCODE genetics/Amgen Inc. The spouse of C.J.W. works at Regeneron. I.S, L.G.F, W.Z., H.L., B.B., J.B.N., M.E.G., A.H.S., B.W., S.E.G., Y.E.C., S.L., H.M.K., A.L., S.F., B.O.Å., and K.H., declare no competing interests.

## Additional information

## Regeneron Genetics Center

Xiaodong Bai[5], Suganthi Balasubramanian[5], Leland Barnard[5], Andrew Blumenfeld[5], Michael Cantor[5], Giovanni Coppola[5], Aris Economides[5], Gisu Eom[5], Lukas Habegger[5], Young Hahn[5], Alicia Hawes[5], Marcus B. Jones[5], Shareef Khalid[5], Luca A. Lotta[5], Evan K. Maxwell[5], Lyndon J. Mitnaul[5], John D. Overton[5], Jeffrey G. Reid[5], Manuel Allen Revez Ferreira[5], William Salerno[5], Deepika Sharma[5], Alan Shuldiner[5], Jeffrey C. Staples[5] & Ashish Yadav[5]

A full list of members appears in the Supplementary Information.

