## [Peer Review File · Nature Communications]

Reviewers' comments:

Reviewer #1 (Remarks to the Author):

This paper reports an exome-sequencing of a subset of the Nord-Trøndelag Health Study with forearm bone mineral density (BMD), followed by imputation in the remaining sample. Following replication in 409,554 UK Biobank participants, they were able to identify 10 loci, including one genome-wide significant loss-of-function (LoF) indel mutation in MEPE gene, associated with decreased BMD and increased osteoporosis and fracture risk. Of note, this insertion/deletion polymorphism was only observed during imputation which included indel calling. This indel seems to be a Norwegian-specific variant, which only to some extent is observed in more heterogeneous UK-whites (18 times less prevalent).

Due to the high impact and effect on BMD observed in mutation carriers of different ages, this finding is of interest (although MEPE gene is well known in the field).

This is a straightforward exome sequencing-informed GWAS, performed by an experienced group, with relatively large numbers in the discovery cohort. The manuscript's methodology is up-to-date and gives evidence of the authors' expertise in the genetic epidemiology. Since the clinical importance of genetic determination of peak bone mass is incontestable, finding "causal" genetic variants is a timely and interesting contribution, which can translate into a clinically-actionable message.

However, the manuscript is not always clear and consistently written. There are several outstanding issues:

- A population-genetic insight would strengthen this paper's message.
- Single-Energy X-Ray Absorptiometry (SXA) - This technique is now less of the focus due to the advent of DXA. In any case, it is used for appendicular skeleton (e.g., radius, wrist, heel) only.
- Replication sample was chosen by convenience, rather than based on phenotype- or ethnic-similarity considerations.

OTHER COMMENTS

Title: a word "lifetime" doesn't fit (this is an adult-only and not a longitudinal study) and should be removed.

Abstract: pls. add "forearm" before bone mineral density (BMD).

Mentioned "cosmopolitan imputation panels" are never discussed in the text. Actually, wouldn't the ethnic nature of the signal argue for use of a population-specific panel instead?

Text:

p. 5: it is unclear whether the "imp r2" and the "imputation quality" is the same measure or different; in any case, pls. use the same number of decimals.

It is unclear why the difference between the carriers of the indel in the UK and Norway is "surprising". This seems to be a Norwegian-specific variant, which only to some extent is reflected in a heterogeneous UK-white population. A similar case was with the study of Styrkarsdottir et al. Nature (2013), who found a LGR4 mutation but couldn't replicate it in a seemingly related sample of Danish women. The authors of this – Norwegian – study might wish to actually try to replicate their signal in the Icelandic sample (based on the common ancestry ties).

On p. 6, the authors "hypothesize that the substantially shortened transcript would be degraded...". This hypothesis is testable, if they have access to the HUNT's homozygotes' stored cells (or iPCs).

p. 9: pls. correct a phrase "forearm BMD measured with DXA" to "SXA".

Methods:

Pls. define HUNT2 and HUNT3 (phases?). Pls. mention how the (Norwegian) exome-sequenced variants were imputed to the rest of the UK Biobank cohort.

The original GWAS from the UK Biobank for eBMD included 426,824 White-British individuals; this study involved 409,554 UK BB participants. Pls. comment on the discrepancy.

MINOR:

"Measurements of the porosity of bones" – neither SXA nor DXA measure "porosity";

"heel calcaneus" – it's either heel or calcaneus;

Pls. re-word: "We identified 10 previously identified loci".

Modeling of the truncated protein using I-TASSER and QUARK algorithms should be mentioned in

the Methods.

"The individuals who carried the p.Lys70IlefsTer26 mutation had a similar rate of fracture" – unclear, similar to whom.

Results: sentence starting with "Importantly" belongs to the Discussion.

Pls. check wording "Ensembl for Refseq" (p. 11).

"genetic principal components" (p. 12) and "principal components of ancestry" (p. 13) seem to be the same. If so, pls. name them consistently.

Figure 2. "lowess smoothing curve" (actually, LOWESS) should be introduced in the Methods.

Table 1. rs753138805 is marked as being "in nearly complete LD with the locus lead SNP" – the latter being, presumably, rs11934731?

Table 2. Are the phenotypes presented in any particular order? What is "Osteoporosis NOS"?

Table 3. pls. remind in the title that the population is HUNT. Pls. remind also, why number of samples with BMD is only 304.

Reviewer #2 (Remarks to the Author):

Surakka et al have done an association in HUNT with WGS and identified a variant in MEPE. It is associated with fracture, osteoporosis diagnosis and BMD. They replicated this in UKB WES and the target UKB imputed dataset. Effects are large and include effects on fracture. The manuscript is generally high quality.

Major issues:

-The authors state in the conclusion of the abstract: "A key implication for precision medicine is that as-yet-undiscovered functional variants not present in the publicly available cosmopolitan imputation panels may be more informative and frequent than polygenic risk scores for some phenotypes." PRSs are continuous scores. Functional variants cannot be more frequent than a continuous score. If that score is dichotomized, then the prevalence of exposure to the functional variant could be higher than that of a dichotomized PRS. So, this interpretation depends on what threshold the PRS is dichotomized at. There is no evidence present in the abstract to defend this far-reaching statement.

-The site that is most often used for a diagnosis of osteoporosis is FN. It is not clear if the diagnosis of osteoporosis was based on a T-score < -2.5 SD from FN. I believe they used ultradistal forearm. This should be clarified throughout the manuscript. It is not clear how T-scores were generated in HUNT for this site. Table 2 shows osteoporosis NOS, but it is not clear what NOS means.

-The main issue with this manuscript is that these results have already been reported by the UKB WES sequencing consortium with replication in HUNT. It is very awkward that the authors have presented here the same variant with identification in HUNT and replication in UKB WES. Some authors overlap between these two papers. I feel that these papers should be merged or published back-to-back in the same issue of a journal. One should not publish the same findings twice, using the same data in my opinion, even if the data are presented in different sequences of discovery and replication in the two manuscripts. In the end, it is the same data that supports the findings.

-The lack of any functional data to support MEPE detracts from the importance of this manuscript.

Minor issues:

- The authors should clarify what is meant in the results by "ultradistal BMD". What site are they referring to? Distal forearm?

- Table 3 should clarify which cohort the data are coming from in the title.

Reviewer #3 (Remarks to the Author):

Nature Communications peer review report

Loss-of-function mutation in the MEPE gene decreases lifetime bone mineral density and increases fracture risk

Surakka et al.

This article describes the discovery of loss of function mutations associated with bone mineral density. The authors performed low coverage sequencing of a subset of participants of a population based study from Norway. This was followed by imputation into the remainder of this cohort, association analysis, and replication using the UK Biobank cohort.

I enjoyed reading the paper. The work is novel, and will be of interest to the scientific community. The paper overall is very well-written. I have suggested minor changes that would improve clarity of the manuscript. The two major comments should be straightforward to address. I recommend acceptance after corrections.

Major comments:

1. In table 2, you selectively report phenotypes of interest from Supplementary table 2. A much better way to do this is to pre-determine a Bonferroni corrected p-value for the 42 phenotypes you have tested ($0.05/42 = 1.2 \times 10^{-3}$), and then report in table 2 all of the phenotypes that passed this corrected p-value threshold. Do not include in Table 2 the phenotypes that didn't pass this threshold.
2. Line 274-277 – What if participants had fractures treated at a hospital in another part of Norway – a trip to Tromsø for example? Why did you restrict to just the local hospitals and not the whole of Norway? If there is no way of looking at the whole of Norway, this should be included in the discussion as a limitation.

Minor comments

Line 99 imputed into n=69716, but the rest of the analysis is on n=19705. This is the subgroup with BMD measures, so I would suggest removing all reference to the 69716 participants throughout the manuscript (and certainly in this paragraph). This will make the paper clearer.

Lines 126, 131 – the SD number is missing

Line 140-148 – no real functional work has been done. This is hypothesis, not results, and I suggest removing this section

Line 222 – Rather than “early initiation”, I think it is possible that carriers could be treated for a short period of life in order to gain better peak BMD, then stop treatment (as BMD loss is at a similar rate to non-carriers). Late childhood / early adulthood may be the important therapeutic window in these people. Please incorporate this concept into the discussion.

Line 232/235 – This long sentence doesn't quite make sense. Please split into two sentences to make the point clearer.

Line 248 – I suggest Table 3 becomes a supplementary table or is removed. I don't think it adds to the message of the paper. Instead, perhaps you could reference a paper describing the HUNT study in detail?

Line 252 – I suggest just describing the imputation into the 19705 with BMD measurements (see above). An alternative is to say in this section that after imputation into the 69716 further analysis was restricted to the 19705 with BMD measurement.

MEPE Response to reviewers Nature Communications

All changes to the manuscript based on Reviewers comments have been highlighted with yellow in the manuscript file and noted below each reply with blue in this document.

Reviewer #1 (Remarks to the Author):

This paper reports an exome-sequencing of a subset of the Nord-Trøndelag Health Study with forearm bone mineral density (BMD), followed by imputation in the remaining sample. Following replication in 409,554 UK Biobank participants, they were able to identify 10 loci, including one genome-wide significant loss-of-function (LoF) indel mutation in MEPE gene, associated with decreased BMD and increased osteoporosis and fracture risk. Of note, this insertion/deletion polymorphism was only observed during imputation which included indel calling. This indel seems to be a Norwegian-specific variant, which only to some extent is observed in more heterogeneous UK-whites (18 times less prevalent).

Due to the high impact and effect on BMD observed in mutation carriers of different ages, this finding is of interest (although MEPE gene is well known in the field).

This is a straightforward exome sequencing-informed GWAS, performed by an experienced group, with relatively large numbers in the discovery cohort. The manuscript's methodology is up-to-date and gives evidence of the authors' expertise in the genetic epidemiology. Since the clinical importance of genetic determination of peak bone mass is incontestable, finding "causal" genetic variants is a timely and interesting contribution, which can translate into a clinically-actionable message.

However, the manuscript is not always clear and consistently written. There are several outstanding issues:

- A population-genetic insight would strengthen this paper's message.

We agree with the Reviewer and thoughts about the population genetic possibilities for the paper as well. Our hypothesis was that the carriers in UK could be carrying the MEPE LoF variant due to the Norwegian/Viking admixture but the further analysis did not give us enough supporting evidence to have it included in the original manuscript

- Single-Energy X-Ray Absorptiometry (SXA) - This technique is now less of the focus due to the advent of DXA. In any case, it is used for appendicular skeleton (e.g., radius, wrist, heel) only.

In HUNT2 all measurements were performed by SXA measuring both ultradistal and distal BMD. In HUNT3 similar equipment was used for about 2/3, while 1/3 was measured by DTX200 which had two x-ray sources and thus was DXA. The latter did only measure distal BMD. The accuracy in the forearm measurement is superior to DXA of lumbar spine due to minimal influence of degenerative changes with calcifications. Even if DXA of the hip is considered as the gold standard of bone densitometry, measurements of the forearm is as good in predicting risk of any fractures. The cut-off for osteoporosis, however, should not be used interchangeably.

Additionally, the Icelandic replication dataset (deCODE) shows similar effect sizes for DXA measured BMDs as HUNT dataset SXA measure assuring that the association we see for the MEPE LoF variant is not due to measurement error.

Changes to the manuscript:

1. Added a short intro to the used measurements into the beginning of the results section (page 5, line 17- page 6, line 4).
2. Added replication from deCODE using DXA measured arm BMD (page 8, lines 1-4 & new Table 2)

- Replication sample was chosen by convenience, rather than based on phenotype- or ethnic-similarity considerations.

To tackle this comment we have contacted our peers in Iceland for more population matching replication. The results of this replication have been added to the results section and new Table 2 and Supplementary Table 3. In addition, we have updated our replication results from UKbb to the new dataset of ~300,000 exomes instead of ~50,000.

Changes to the manuscript:

1. Added deCODE replication and methods (4 new authors [U.T., H.H., D.G., K.S.] , abstract, lines 8-9 & page 5, lines 12-13 & page 8, lines 1-4 & new Table 2 & page 8, lines 11-13 & new Supplementary Table 3 & page 16, lines 1-7 & edited Disclosures page 16, line 12 & edited acknowledgements page 16, line 17 & added new references #30)
2. Updated UK Biobank replication results and methods (one new author [J.A.K.] & abstract, lines 8-9 & page 7, line 19 -page 8, line 1 & page 8, lines 13-14 & page 15, lines 12-23)

OTHER COMMENTS

Title: a word "lifetime" doesn't fit (this is an adult-only and not a longitudinal study) and should be removed.

The title has been changed to better reflect the nature of the discovery dataset

Changes to the manuscript:

Changed "lifetime" to "ultradistal forearm" in the Title.

Abstract: pls. add "forearm" before bone mineral density (BMD).

Added

Changes to the manuscript:

Changed "lifetime" to "ultradistal forearm" in the Title.

Mentioned "cosmopolitan imputation panels" are never discussed in the text. Actually, wouldn't the ethnic nature of the signal argue for use of a population-specific panel instead?

We wholeheartedly agree and had used a study design that included low-pass whole genome sequencing of 2,202 genomes, including calling indels, which was combined with the HRC panel (removing overlapping HUNT samples which we contributed to the HRC), and imputed into our dataset. We have explained this approach in the results and methods sections to make our design more clear. The sentence where we refer to the cosmopolitan panels, is to hypothesize the overall lack of LoF variant discoveries being possibly due to the missing population specific variation. We have fixed this sentence in the abstract to make our message clearer

Changes to the manuscript:

Re-worded abstract (abstract, lines 11-13)

Text:

p. 5: it is unclear whether the "imp r2" and the "imputation quality" is the same measure or different; in any case, pls. use the same number of decimals.

It is the same measure, we clarified the text to use a standard naming scheme (imputation R^2).

Changes to the manuscript:

1. Changed "imputation r2" to "imputation R2" on page 6, line 23.
2. Changed "imputation quality" to "imputation R2" on page 7, line 2

It is unclear why the difference between the carriers of the indel in the UK and Norway is "surprising". This seems to be a Norwegian-specific variant, which only to some extent is reflected in a heterogeneous UK-white population. A similar case was with the study of Styrkarsdottir et al. Nature (2013), who found a LGR4 mutation but couldn't replicate it in a seemingly related sample of Danish women. The authors of this – Norwegian – study might wish to actually try to replicate their signal in the Icelandic sample (based on the common ancestry ties).

Reviewers at a previous journal disliked us calling this a Norwegian-specific variant because it was identified in the British population. We attempted to use this variant to trace potential Norwegian/Viking admixture into the British population but could find no definitive evidence. We have changed the wording to remove the 'surprising' comment.

Changes to the manuscript:

Removed "Surprisingly, " from page 7, line 22 (Sentence used to start "Surprisingly, the proportion of UK individuals")

On p. 6, the authors "hypothesize that the substantially shortened transcript would be degraded...". This hypothesis is testable, if they have access to the HUNT's homozygotes' stored cells (or iPSCs).

Unfortunately, we do not have access to iPSC or stored cells from HUNT homozygotes. We searched for homozygotes in our local biobank also but none existed.

p. 9: pls. correct a phrase "forearm BMD measured with DXA" to "SXA".

Corrected

Changes to the manuscript:

Changed "DXA" to "SXA" on page 11, line 22

Methods:

Pls. define HUNT2 and HUNT3 (phases?). Pls. mention how the (Norwegian) exome-sequenced variants were imputed to the rest of the UK Biobank cohort.

HUNT biobank collects data about every ten years. HUNT2 and HUNT3 are two separate time windows for the data collection. We have clarified this in the Methods section.

UKbb used their own imputation panel of ~50,000 exome sequences samples, as is described in their pre-printed manuscript (Reference 24). However, as our collaborators at Regeneron have exome sequenced additional samples, instead of using the imputed variant we have now changed the UKbb replication results to sequenced samples only (new N~300,000).

Changes to the manuscript:

1. Added a short intro to the used measurements into the beginning of the results section (page 5, line 17- page 6, line 4).
2. Clarified the methods section on page 13, lines 7-8 & page 14, lines 3-4

The original GWAS from the UK Biobank for eBMD included 426,824 White-British individuals; this study involved 409,554 UK BB participants. Pls. comment on the discrepancy.

The original BMD analysis in UKbb (the UKbb exome sequence manuscript, Reference 14) was done using a composite definition that excluded some samples. However, the new results from the 300,000 exome sequenced samples has now been done using the same eBMD definition as Morris et al. Nat Gen 2019 for clarity and added a reference to this paper to the UKbb methods section.

Changes to the manuscript:

1. New replication results from UK Biobank (page 7, line 19 - page 8, line 1 & page 8, lines 13-14 & page 15, lines 12-23)
2. Reference to Morris et al (ref #10). on page 15, line 18

MINOR:

"Measurements of the porosity of bones" – neither SXA nor DXA measure "porosity";

Changed "porosity of bones" to "density of bones" on page 4, line 4

"heel calcaneus" – it's either heel or calcaneus;

Removed "calcaneus" on page 4, line 16

Pls. re-word: "We identified 10 previously identified loci".

Changed to "We replicated 10 previously identified loci" on page 6, lines 9-10

Modeling of the truncated protein using I-TASSER and QUARK algorithms should be mentioned in the Methods.

We have removed this section by suggestion of Reviewer #3

"The individuals who carried the p.Lys70IlefsTer26 mutation had a similar rate of fracture" – unclear, similar to whom.

Corrected on page 8, line 23 - page 9, line 3

Results: sentence starting with "Importantly" belongs to the Discussion.

We have moved this sentence to the Discussion section Page 11, lines 7-10 & lines 14-16

Pls. check wording "Ensembl for Refseq" (p. 11).

Corrected "for" to "or" and capitalized the "S" in RefSeq on page 13, line 14

"genetic principal components" (p. 12) and "principal components of ancestry" (p. 13) seem to be the same. If so, pls. name them consistently.

Changed "principal components of ancestry" to "genetic principal components" on page 15, line 21

Figure 2. "lowess smoothing curve" (actually, LOWESS) should be introduced in the Methods.

We added a paragraph into the Methods section describing this method (page 15, lines 7-9) and capitalized the name of the method in the Figure 2 legend.

Table 1. rs753138805 is marked as being "in nearly complete LD with the locus lead SNP" – the latter being, presumably, rs11934731?

We are referring to the lead SNP of the locus in our dataset, rs181831514. We have clarified this in the text (page 6, line 16). Additionally, we have made a new Table 2 showing the results for the LoF variant separately.

Table 2. Are the phenotypes presented in any particular order? What is "Osteoporosis NOS"?

This is the new Table 3. We have re-ordered the rows by P-value to be clearer and only reported variants that reach Bonferroni corrected P-value (request from Reviewer #3). We added the definition of "NOS" to the Supplementary Table 2 footnote.

Table 3. pls. remind in the title that the population is HUNT. Pls. remind also, why number of samples with BMD is only 304.

We have added the source of the data to the Table title (New Supplementary Table 6 by the request of Reviewer #3). As the table is showing clinical characteristics for individuals carrying different *MEPE* mutations compared to the whole dataset, the 304 are the individuals carrying the *MEPE* LoF for whom we also have the BMD measurement.

Reviewer #2 (Remarks to the Author):

Surakka et al have done an association in HUNT with WGS and identified a variant in *MEPE*. It is associated with fracture, osteoporosis diagnosis and BMD. They replicated this in UKB WES and the largest UKB imputed dataset. Effects are large and include effects on fracture. The manuscript is generally high quality.

Thank you

Major issues:

-The authors state in the conclusion of the abstract: "A key implication for precision medicine is that as-yet-undiscovered functional variants not present in the publicly available cosmopolitan imputation panels may be more informative and frequent than polygenic risk scores for some phenotypes." PRSs are continuous scores. Functional variants cannot be more frequent than a continuous score. If that score is dichotomized, then the prevalence of exposure to the functional variant could be higher than that of a dichotomized PRS. So, this interpretation depends on what threshold the PRS is dichotomized at. There is no evidence present in the abstract to defend this far-reaching statement.

Thank you, fair point. We were comparing to the effect of one standard deviation change in the PRS, but we have clarified this in the text by adding one more comparison point 5% on top of the presented 1% and 10% in Supplementary Table 4. Additionally, we have added a sentence to the results of the other comparison points and re-wrote the abstract to better reflect our conclusion.

Changes to the manuscript:

Edited abstract lines 11-13, added results from other comparison points to page 9, lines 10-13, added 5% cut-point results to Supplementary Table 4.

-The site that is most often used for a diagnosis of osteoporosis is FN. It is not clear if the diagnosis of osteoporosis was based on a T-score < -2.5 SD from FN. I believe they used ultradistal forearm. This should be clarified throughout the manuscript. It is not clear how T-scores were generated in HUNT for this site. Table 2 shows osteoporosis NOS, but it is not clear what NOS means.

As we don't have the official T-score measurement available, which is always in relation to a reference dataset BMD distribution, we have used within dataset -2.5 SD units of the forearm BMD as the closest proxy. In the text we had tried to make the distinction by always referring to 2.5 SD instead of T-score but have now clarified this more by adding a sentence "As we do not have the official T-score available for the dataset, which is always in relation to a reference dataset BMD distribution, we are using the forearm BMD < -2.5 SD as a proxy."

We have added the definition of NOS to the Supplementary Table 2 footnote which is the only appearance of the abbreviation after removing rows not reaching Bonferroni corrected P-value from the Table 2 (as was suggested by Reviewer #3).

Changes to the manuscript:

Added more detailed description of the measurement on page 8, lines 16-18.

Added a short description of the used traits to the beginning of the results section (page 5, line 17- page 6, line 4)

Added “ultradistal forearm” to multiple occurrences of “BMD” in the text (Title & abstract line 7 & page 7, line 15 & page 8, line 21-22 & page 9, lines 14&18&19 & page 10, line 3 & page 11, line 18 & page 14, line 22 & Figure 1 legend)

-The main issue with this manuscript is that these results have already been reported by the UKB WES sequencing consortium with replication in HUNT. It is very awkward that the authors have presented here the same variant with identification in HUNT and replication in UKB WES. Some authors overlap between these two papers. I feel that these papers should be merged or published back-to-back in the same issue of a journal. One should not publish the same findings twice, using the same data in my opinion, even if the data are presented in different sequences of discovery and replication in the two manuscripts. In the end, it is the same data that supports the findings. The results have not been published, only deposited in a non-peer reviewed pre-print archive (bioRxiv). The methods for discovery are very different and our manuscript dives deeper into the phenotype, including association with fractures. The UK biobank whole exome sequencing manuscript is focused on phenome and exome-wide discovery and genetic architecture. Because the variant is present at much lower frequency in the UK, there is insufficient power to detect association with fracture in the UK biobank data. Michelle Trenkmann, editor at Nature Communications, is willing to work with the editor at Nature (where the UKB WES manuscript will soon be resubmitted), to coordinate publication. Science works best when results are independently replicated, and in this case, the discovery was completely independent but known to both teams because of Goncalo Abecasis's move from the University of Michigan HUNT analysis team to Regeneron. The HUNT study discovered this association a full year prior to Regeneron (and indeed prior to Regeneron having any exome sequencing data) but spent time attempting (and failing) to generate functional data to accompany the manuscript.

-The lack of any functional data to support MEPE detracts from the importance of this manuscript. We attempted to develop functional results by examining the gene in osteoblasts but the results were inconclusive due to technical failure.

Minor issues:

- The authors should clarify what is meant in the results by “ultradistal BMD”. What site are they referring to? Distal forearm?

Yes, the BMD has been measured from the ultradistal part of the non-dominant forearm. To be more clear we have added “ultradistal forearm” to multiple occurrences of “BMD” in the text (Title & abstract line 7 & page 7, line 15 & page 8, line 21-22 & page 9, lines 14&18&19 & page 10, line 3 & page 11, line 18 & page 14, line 22 & Figure 1 legend)

- Table 3 should clarify which cohort the data are coming from in the title.

We will, thank you. The HUNT study leaders will be very happy. The Table has been moved to the supplements (new Supplementary Table 6) by the request of Reviewer #3

Changes to the manuscript:

Added the name of the cohort (HUNT) to the Supplementary Table 6 title

Reviewer #3 (Remarks to the Author):

Nature Communications peer review report

Loss-of-function mutation in the MEPE gene decreases lifetime bone mineral density and increases fracture risk
Surakka et al.

This article describes the discovery of loss of function mutations associated with bone mineral density. The authors performed low coverage sequencing of a subset of participants of a population based study from Norway. This was followed by imputation into the remainder of this cohort, association analysis, and replication using the UK Biobank cohort.

I enjoyed reading the paper. The work is novel, and will be of interest to the scientific community. The paper overall is very well-written. I have suggested minor changes that would improve clarity of the manuscript. The two major comments should be straightforward to address. I recommend acceptance after corrections.

Thank you

Major comments:

1. In table 2, you selectively report phenotypes of interest from Supplementary table 2. A much better way to do this is to pre-determine a Bonferroni corrected p-value for the 42 phenotypes you have tested ($0.05/42 = 1.2 \times 10^{-3}$), and then report in table 2 all of the phenotypes that passed this corrected p-value threshold. Do not include in Table 2 the phenotypes that didn't pass this threshold.

Thank you, we have done this.

Changes to the manuscript:

1. Removed lines from Table 3 (old Table 2) based on the suggestion
2. Added a sentence to the methods describing the used Bonferroni correction (page 15, lines 6-7)

2. Line 274-277 – What if participants had fractures treated at a hospital in another part of Norway – a trip to Tromsø for example? Why did you restrict to just the local hospitals and not the whole of Norway? If there is no way of looking at the whole of Norway, this should be included in the discussion as a limitation.

This data is not available to the HUNT study, as there is no nation-wide registry that covers fractures for the entire follow-up period of our study. We have now included this as a limitation. Nonetheless, any such incompleteness in the recording of fracture outcomes is likely small and is unlikely to differ by MEPE variants, and is therefore unlikely to have substantially biased our results.

Changes to the manuscript:

Added a sentence to the discussion (page 12, lines 4-6)

Minor comments

Line 99 imputed into $n=69716$, but the rest of the analysis is on $n=19705$. This is the subgroup with BMD measures, so I would suggest removing all reference to the 69716 participants throughout the manuscript (and certainly in this paragraph). This will make the paper clearer.

Good suggestion, thank you. We have now clarified that the BMD association has been tested in 19,705 samples whereas the clinical end-point association analyses have been tested in the full dataset ($N=69,716$)

Changes to the manuscript:

1. Added sample size of BMD analysis to page 6, line 9
2. Added sample size of clinical end-point analysis to the page 8, lines 8-9
3. Added word "discovery" on page 14, line 5
4. Edited a sentence on page 15, lines 5-6

Lines 126, 131 – the SD number is missing

The SD here stands for SD units for the effect size. We have clarified this notation by adding clear definition to the first occurrence of the abbreviation.

Changes to the manuscript:

1. Added "[standard deviation units]" to page 6, line 17
2. Added description for the effect size to the Table 1 footnote

Line 140-148 – no real functional work has been done. This is hypothesis, not results, and I suggest removing this section

We will, thank you.

Changes to the manuscript:

Removed section "MEPE Loss-of-function deletion"

Line 222 – Rather than “early initiation”, I think it is possible that carriers could be treated for a short period of life in order to gain better peak BMD, then stop treatment (as BMD loss is at a similar rate to non-carriers). Late childhood / early adulthood may be the important therapeutic window in these people. Please incorporate this concept into the discussion.

We have edited the Discussion section by this insightful suggestion

Changes to the manuscript:

Added sentence to the discussion page 11, lines 8-10

Line 232/235 – This long sentence doesn't quite make sense. Please split into two sentences to make the point clearer.

We have divided this sentence into two to make it more clear

Changes in the manuscript:

Two sentences instead of one on page 12, lines 11-15

Line 248 – I suggest Table 3 becomes a supplementary table or is removed. I don't think it adds to the message of the paper. Instead, perhaps you could reference a paper describing the HUNT study in detail?

Thank you, we have now moved the table to supplements (new Supplementary Table 6). In the manuscript, we have a reference for the HUNT study, ref#11, and for the BMD measurements, ref#28

Line 252 – I suggest just describing the imputation into the 19705 with BMD measurements (see above). An alternative is to say in this section that after imputation into the 69716 further analysis was restricted to the 19705 with BMD measurement.

Thank you, we have now tried to clarify the sample sizes in the analysis throughout the manuscript

Changes to the manuscript:

1. Added sample size of BMD analysis to page 6, line 9
2. Added sample size of clinical end-point analysis to the page 8, lines 8-9
3. Added word “discovery” on page 14, line 5
4. Edited a sentence on page 15, lines 5-6

Dominic Furniss
Oxford University, UK

REVIEWERS' COMMENTS:

Reviewer #1 (Remarks to the Author):

The authors have provided appropriate responses to all the concerns this reviewer raised. The manuscript is now more clearly and consistently written. Still, there are several omissions or ambiguous expressions, for the authors to correct:

Line 58: pls. re-order the sentence "Due to the observed lifelong effect on BMD, ...".

II. 277-278: sentence "Approximately 70,000 HUNT individuals have been genotyped ..." – is unclear; probably, a word is missing.

I. 347: add "arm" to the deCODE's BMD (as of now, it has just lumbar spine, hip and whole body).

The abbreviation (DXA) should be defined at its first occurrence.

Reviewer #2 (Remarks to the Author):

The authors have addressed my comments. Thank you

Reviewer #3 (Remarks to the Author):

I have reviewed the revised manuscript. The authors have incorporated the changes and suggestions I have made, and most of those from other reviewers. I think the manuscript is improved, and I would recommend publication.

RESPONSE TO REVIEWERS:

REVIEWERS' COMMENTS:

Reviewer #1 (Remarks to the Author):

The authors have provided appropriate responses to all the concerns this reviewer raised. The manuscript is now more clearly and consistently written. Still, there are several omissions or ambiguous expressions, for the authors to correct:

Line 58: pls. re-order the sentence "Due to the observed lifelong effect on BMD, ...".

We have now changed this sentence to be more clear

II. 277-278: sentence "Approximately 70,000 HUNT individuals have been genotyped ..." – is unclear; probably, a word is missing.

We have reworded this sentence to clarify

I. 347: add "arm" to the deCODE's BMD (as of now, it has just lumbar spine, hip and whole body).

The arm has been added to the sites of measurement

The abbreviation (DXA) should be defined at its first occurrence.

This has now been added

Reviewer #2 (Remarks to the Author):

The authors have addressed my comments. Thank you

Reviewer #3 (Remarks to the Author):

I have reviewed the revised manuscript. The authors have incorporated the changes and suggestions I have made, and most of those from other reviewers. I think the manuscript is improved, and I would recommend publication.